# Reliability of Noninvasive Sonic Tomography for the Detection of Internal Defects in Old, Large Trees of *Abies holophylla* Maxim

**Jiwon Son *** , **Gwanggyu Lee and Jinho Shin**

Division of Natural Heritage, National Research Institute of Cultural Heritage, 927 Yudeung-ro, Seo-gu, Daejeon 34122, Korea; y2376el@korea.kr (G.L.); Sjho77@korea.kr (J.S.)
* Correspondence: wine814@korea.kr

**Abstract:** Internal decay and cavities in wood are known to reduce the structural functionality of trees. Such damage may lead to detrimental effects not only on the wood, but also on humans. This is especially the case with old, large trees that are more vulnerable to heavy snow and strong wind. Thus, preventative management (e.g., detecting internal wood defects) is essential. The present study investigated the reliability of noninvasive measurements using sonic tomography (SoT) to detect internal defects in *Abies holophylla* Maxim. trees and compared the results with measurements using the invasive method of resistance microdrilling (RM). The tomograms were visually compared with tree cross-section images. The results of SoT and RM showed no significant differences, while the explanatory power, as determined by a regression analysis, were considerably high at 67% with a positive correlation between the two methods. In comparison to the cross-section images, the tomograms were found to reflect the size and position of internal decay, although the detected size tended to be larger than the actual decay area. Our findings indicate SoT as a promising noninvasive technique for detecting internal defects in *A. holophylla* trees.

**Keywords:** noninvasive measurement; resistance microdrilling; acoustic tomography; PICUS3 sonic tomography

## 1. Introduction

The loss of structural functionality in trees due to internal wood decay or cavities may decrease the value of the trees and may even cause severe damage to humans or property upon heavy snow and strong wind. Notably, trees growing in cultural heritage sites, such as a national historic site, or trees designated as important to cultural heritage, such as natural monuments, tend to be of an older age and are, thus, more vulnerable to strong weather and reductions in structural stability. Furthermore, trees that are deemed to have heritage value are important to the general public. It is, thus, important to prevent physical damage through preventive management based on the detection of internal defects. However, as it is difficult to visually discern internal decay or structural defects, a noninvasive method to detect such defects is necessary.

In the past two decades, studies have actively investigated the use of noninvasive techniques to detect internal wood decay or cavities. These techniques include X-ray computerized tomography [1], stress waves [2], and ultrasound [3,4]. The present study applied sonic tomography (SoT) to detect internal wood defects. The potential of SoT as a method to detect internal defects in old, large trees is high, as it allows the collection of data regarding the cross section of an entire tree as well as measurements of large trees through stable field application. Studies on noninvasive measurements of trees have mainly been conducted on European and tropical tree species [5–12], while studies in Asia have, for example, analyzed the reliability of SoT for measuring naturally growing trees in city parks in China and *Zelkova serrata* Makino [13,14]. Few studies have reported on naturally growing trees on the Korean peninsula or in other Asian countries. SoT should first be

verified for reliability in relation to different tree species, as the measurement accuracy may vary according to species-specific wood characteristics.

To verify its reliability, SoT was compared with the invasive method resistance microdrilling (RM) for detecting decay within trees of *Abies holophylla* Maxim. Resulting tomograms were compared with cross-section images after felling. *Abies holophylla* naturally grows in South Korea, China, and eastern regions of Russia under the influence of the East Asian continental climate. Various species have been found across the Northern Hemisphere. Nonetheless, *A. holophylla* requires conservation as it is included as a "Near Threatened" species in the red list of the International Union for Conservation of Nature (IUCN). Notably, *A. holophylla* in South Korea has been designated as a natural monument as it is the main traditional landscape species in South Korean temple forests. In 2019, an *A. holophylla* tree of approximately 250 years of age, a natural monument in the Haksadae Pavilion, Hapcheon, South Korea, fell due to a typhoon, and a large internal cavity that was difficult to visually detect was found within the fallen tree. As such, a noninvasive method for detecting such internal defects is essential, especially in old, large trees of national importance, as it is highly difficult to visually detect such defects. Noninvasive measurements are anticipated to allow the preventive management of such trees to minimize damage.

In the present study, tomograms of old, large trees of *A. holophylla* from noninvasive SoT and invasive RM were cross-compared, and the tomograms were also compared with cross-section images of the trees. The aim was to verify the reliability of SoT so that the technique may be applied for detecting defects in *A. holophylla* in the future in a noninvasive manner.

## 2. Materials and Methods

A field investigation was conducted in June 2021 to analyze six trees of *A. holophylla* aged 80–100 years in Gwangneung town and Gwangneung forest in Pocheon city, Gyeonggi-do, and Yeongneung, Yeoju-si, Gyeonggi-do, in South Korea. At these locations, *A. holophylla* represents old, large trees growing in a historic site alongside the tombs of the fourth King of Joseon Dynasty, King Sejong, and the Queen, and of the seventh King of Joseon Dynasty, King Sejo, and the Queen. Gwangneung forest, in particular, was designated as a culturally important forest to protect the tombs during the era of King Sejo, and as it is also designated as a UNESCO Biosphere Reserve, the forest is a rare natural site with historical as well as natural value.

For the detection of internal defects in *A. holophylla*, SoT was performed as a noninvasive measurement. To verify the reliability of the SoT results, SoT and RM were performed on the same 54 measuring points (MPs) across six trees. For the cross-validation of SoT and RM measurements, the defect length (cm) was compared among the 54 MPs. Using SPSS 21.0 (IBM Corp., Armonk, NY, USA), independent t-test and simple linear regression were performed. When measurements were completed, three trees were cut to collect five disks, and the defects (decay, cavities, and cracks) on the cross-section images were compared with the tomography results.

In SoT, the transmission speed of stress waves is measured from various directions using a sensor, and the speed varies according to wood elasticity and density. In general, internal wood decay or cavities show a decreasing trend in sonic speed in comparison to healthy areas [11]. The present study used PiCUS 3 Sonic Tomography (Argus Electronic GmbH, Rostock, Germany) to measure daytime cross sections 25–100 cm from the ground. The PiCUS 3 software converts sonic speed into a 2D sonic tomogram in which dark brown indicates high sonic speed and healthy wood (high density), red and blue indicate slow sonic speed and a defect (e.g., decay or cavity), and green indicates an early fungal infection, i.e., a transition between healthy and unhealthy wood.

At the end of the SoT, the trees were cut, and electric resistance tomography (ERT) was performed to detect the areas of early decay that are difficult to observe using SoT. ERT is based on the principle of measuring the percentage of water content and the electrical

resistance in tree cross sections based on the current and voltage. The device used in this study was the PiCUS 3 Treetronic (Argus Electronic).

To collect comparable data for verifying the reliability of SoT results, the microinvasive technique RM (Resistograph 650, Rinntech, Heidelberg, Germany) was applied. In RM, a drill needle (tip diameter 3 mm and shaft diameter 1.5 mm) passes through wood sections of different density to detect internal defects, with high accuracy, based on the measured changes in density. The method is thus widely used for the detection of wood decay or structural diagnosis [15–18]. However, RM has limitations in that it is influenced by water content and leaves small holes in the wood, with potential wood infection and discoloration due to the drilling. In the present study, the RM measurements were taken from 54 MPs based on the areas suspected of defect on tomograms and four bearings from the bark of the tree towards the pith (radial direction), in parallel to the ground surface.

To compute the area of defective portions for each tree from the SoT and wood cross sections, the areas were quantified through image analysis using ImageJ software (National Institutes of Health; http://imagej.nih.gov (15 August 2021, open source).

## 3. Results

### 3.1. Cross-Comparison of SoT and RM Measurements

The results of RM and SoT measurements are presented in Table 1. To verify the reliability of the results, the RM was performed on the same 54 MPs as SoT, and the collected data were analyzed by independent t-test and regression. Figure 1 illustrates the SoT and RM results. The red line indicates the depth of the internal defect (e.g., decay or cavity), and green indicates sound wood in RM results. The defective section (blue) on the tomogram of each of the 54 MPs and the defect lengths measured by RM showed a deviation of approximately 5.6 cm on average. Based on the t-test, the defective area (blue) on the tomogram was 13.2 cm on average, and the defect measured by RM was 11.9 cm on average, with no significant intergroup difference ($t = 0.437$, $p = 0.663$) (Table 1).

**Table 1.** Results of *t*-test on lengths of defects measured with sonic tomography (SoT) and resistance microdrilling ($N = 54$). MP: measuring point; SoT blue: blue and red color in the SoT tomograms.

| MP | Defect Length (cm) | | | Measuring Length (cm) | MP | Defect Length (cm) | | | Measuring Length (cm) |
|---|---|---|---|---|---|---|---|---|---|
| | SoT Blue (A) | Resistance Microdrilling (B) | Difference (A − B) | | | SoT Blue (A) | Resistance Microdrilling (B) | Difference (A − B) | |
| 1 | 0 | 0 | 0 | 50 | 1 | 0 | 0 | 0 | 50 |
| 9 | 0 | 0 | 0 | 50 | 15 | 12 | 0 | 12 | 50 |
| 6 | 0 | 0 | 0 | 50 | 7, 8 | 50 | 47.5 | 2.5 | 50 |
| 3 | 0 | 0 | 0 | 50 | 2 | 32.6 | 15.2 | 17.4 | 50 |
| 3 | 0 | 0 | 0 | 47.5 | 14 | 27 | 20 | 7 | 50 |
| 10 | 0 | 0 | 0 | 47.5 | 9 | 46 | 44.1 | 1.9 | 50 |
| 8 | 0 | 0 | 0 | 47.5 | 2 | 27.4 | 24 | 3.4 | 49.5 |
| 5, 6 | 0 | 0 | 0 | 47.5 | 14 | 22 | 22.5 | 0.5 | 50 |
| 5 | 26.7 | 33.5 | 6.8 | 50 | 9 | 45 | 45 | 0 | 50 |
| 7 | 30 | 29 | 1 | 50 | 1 | 1.5 | 5 | 3.5 | 47.5 |
| 9 | 28 | 38.5 | 10.5 | 50 | 4, 5 | 1.5 | 1 | 0.5 | 42.8 |
| 2 | 24.6 | 29 | 4.4 | 50 | 7 | 5 | 1 | 4 | 50 |
| 1 | 21.4 | 37.5 | 16.1 | 50 | 11 | 0 | 9.5 | 9.5 | 50 |
| 6, 7 | 44.3 | 6 | 38.3 | 50 | 4 | 0.9 | 4.5 | 3.6 | 50 |
| 7, 8 | 30.3 | 23 | 7.3 | 50 | 12 | 0 | 2.5 | 2.5 | 50 |
| 5, 6 | 35 | 17 | 18 | 50 | 2 | 5.7 | 1 | 4.7 | 45.8 |
| 1, 2 | 0 | 6.5 | 6.5 | 50 | 6 | 3.3 | 11.5 | 8.2 | 50 |
| 15, 16 | 9.4 | 5 | 4.4 | 50 | 2 | 0 | 0 | 0 | 50 |
| 10 | 39 | 22.3 | 16.7 | 50 | 4 | 0 | 4 | 4 | 50 |
| 10 | 8.8 | 11 | 2.2 | 50 | 12 | 0 | 0.5 | 0.5 | 50 |
| 7, 8 | 32 | 27.5 | 4.5 | 50 | 10 | 0 | 7.5 | 7.5 | 40.3 |
| 8 | 16.5 | 16.5 | 0 | 50 | 10 | 0 | 2.5 | 2.5 | 38.3 |
| 5, 6 | 30 | 10.5 | 19.5 | 50 | 6 | 0 | 2 | 2 | 50 |
| 1 | 0 | 0 | 0 | 50 | 10, 11 | 0 | 1.3 | 1.3 | 50 |
| 15 | 0 | 0 | 0 | 50 | 1 | 0 | 15.6 | 15.6 | 24.1 |
| 7, 8 | 27 | 25.5 | 1.5 | 50 | 4 | 0 | 5 | 5 | 23.9 |
| 5, 6 | 30.8 | 15 | 15.8 | 50 | 9 | 0 | 10.4 | 10.4 | 20.9 |

| Category | Mean | | Standard Deviation | | *t* | *p* |
|---|---|---|---|---|---|---|
| | Resistance Microdrilling | SoT (Blue) | Resistance Microdrilling | SoT (Blue) | | |
| Defect Length | 11.9 | 13.2 | 13.7 | 16 | 0.437 | 0.663 |

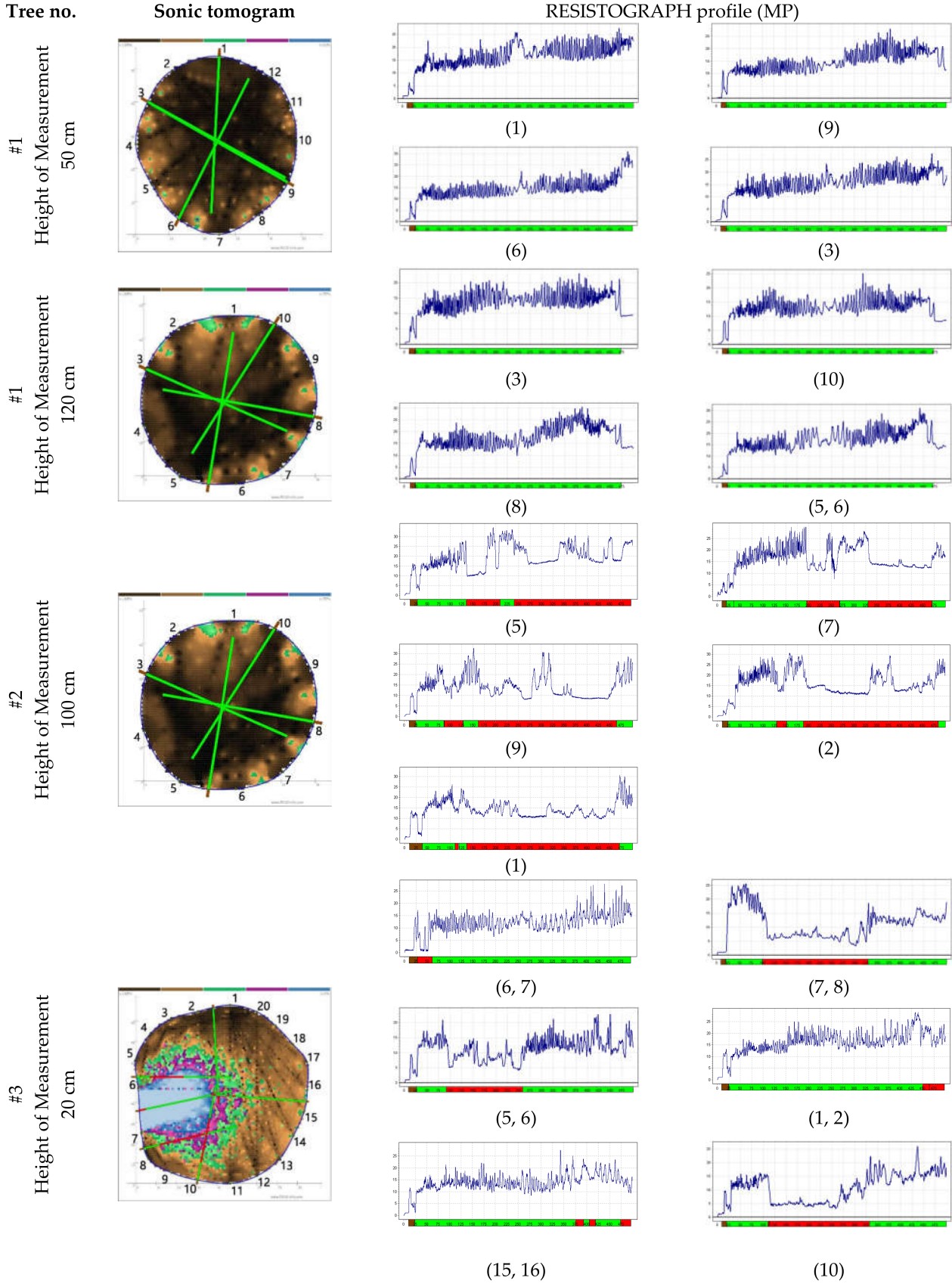

**Figure 1.** *Cont.*

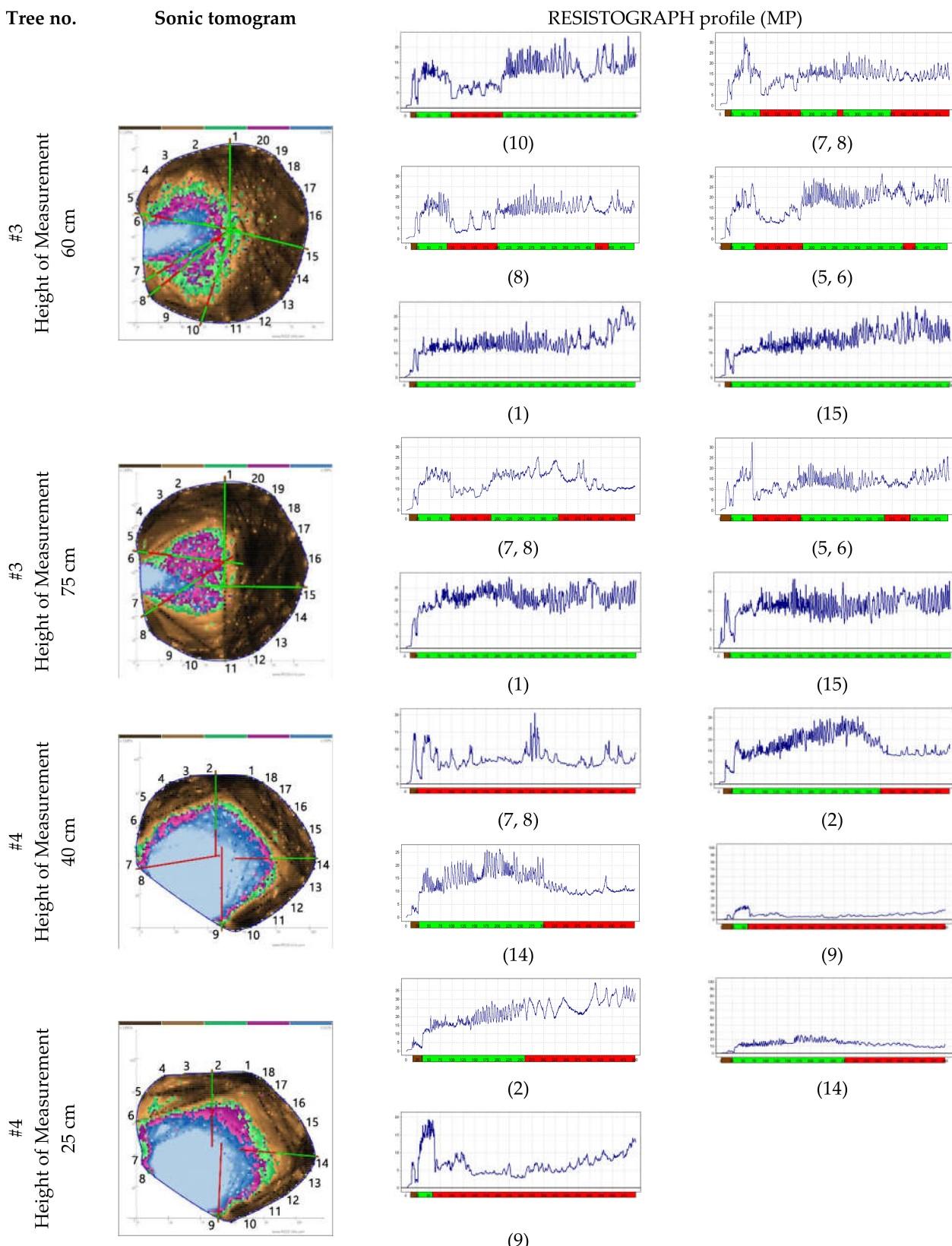

**Figure 1.** *Cont.*

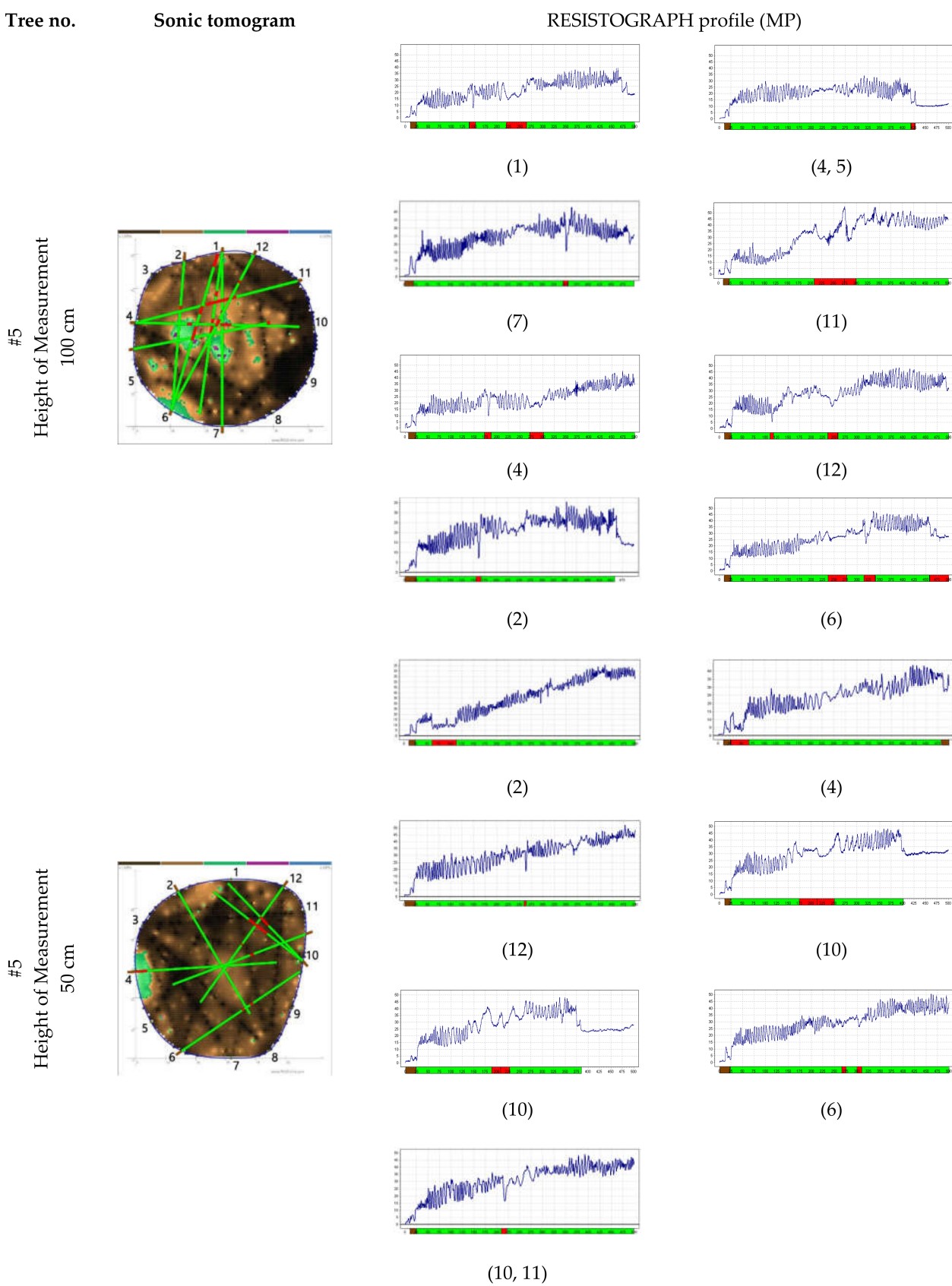

**Figure 1.** *Cont.*

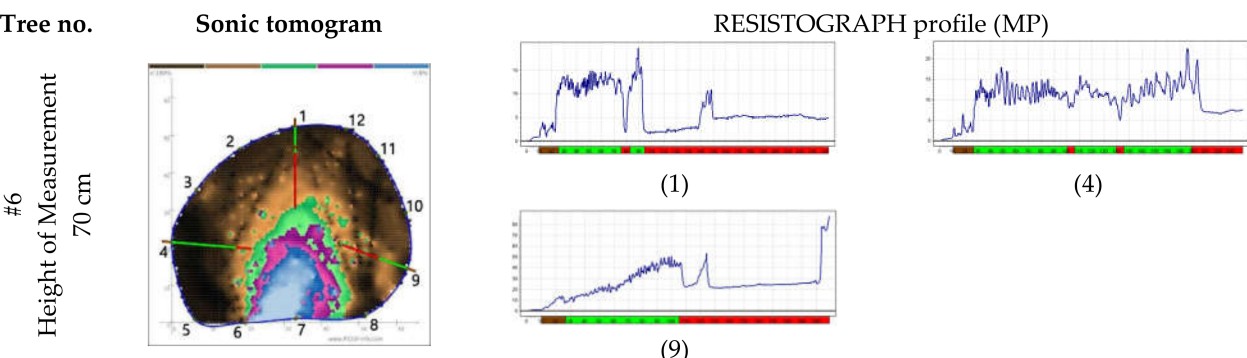

**Figure 1.** Sonic tomograms superimposed with the RESISTOGRAPH profile for selected drilling measurement points (MPs). In the sonic tomograms and resistance profiles graphs, green lines indicate the depth of the drilling path and sound region, red lines represent the depth of the decay region, and brown lines represent bark. In the sonic tomograms, brown: healthy woods, red and blue: defects, green: initial decays.

The regression analysis for determining the correlations between the two measurement results showed a significant regression model with 67% explanatory power for the changes in RM values with $R^2 = 0.675$ (F = 108.214, $p < 0.001$; revised $R^2 = 0.669$). The defect lengths by SoT and RM were found to be significantly positively correlated, indicating that an increase in defects detected by SoT meant an increase in defects measured by RM (Figure 2). This suggested that SoT was a reliable method to detect internal wood defects in *A. holophylla* trees.

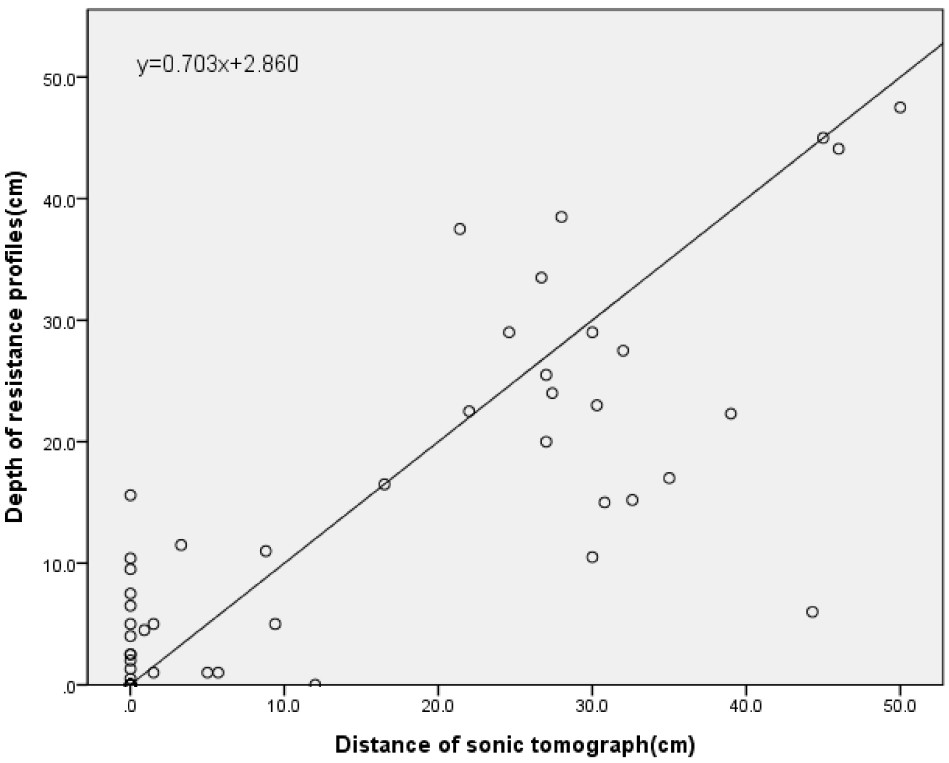

**Figure 2.** Predicted distance of compromised wood using sonic tomography versus the distance of defects using resistography. Trend line = linear regression. $p < 0.001$, $r^2 = 0.675$, $n = 54$.

### 3.2. Comparison of Sonic Tomograms and Cross-Section Images

The SoT results for the five disk samples from three cut trees of *A. holophylla* are presented in Figure 3. In the tomogram shown in Figure 3a, the decay area is presumed to be increasing towards the pith at 6 and 7 MPs. In the disk image, likewise, the decayed section (red dotted line) is shown to be approaching the pith at 6 and 7 MPs (open cavity).

A similar result is shown in Figure 3b,c, where the defect is expanding toward the center from the rings at 8 and 9 MPs in the tomogram. The area was estimated to be a decayed section based on the high percentage of water content in the ERT. As the decay can be seen at the same location on the disk, the tomogram may be taken to indicate the actual defect with similarities, but the estimated values indicated the detection of a larger area of defect. Figure 3d,e show sound wood with some cracks (10 cm in length), a knot, and discoloration. No crack was detected in the tomogram, while such a crack was detected in the RM profile. As a result of quantitatively analyzing the defective area using ImageJ, the error between the defective area by SoT and the actual defective area of the wood section was found to be ±11.3% (Table 2). Specifically in Table 2a, the defective area detected by SoT was 26.5%, whereas the actual defective area was 38.8%, indicating an error of about 12.3%. In Table 2b,c showing the cross section of tree no. 4, an error of about 21–22% was identified. For the cross section of tree no. 4, SoT overestimated the defective area. Figure 3d,e show sound wood with some cracks (10 cm in length), a knot, and discoloration. These results were almost the same as those for SoT, indicating a sound internal wood condition, with few or no internal defects. However, no crack was detected in the tomogram, while cracks were detected in the RM profile.

**(a) Tree no.3: Height of Measurement 20 cm**

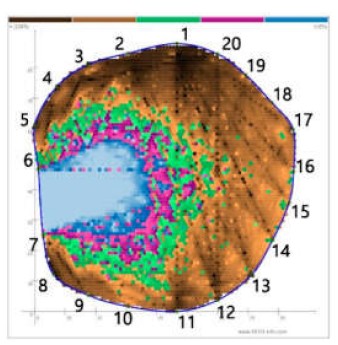 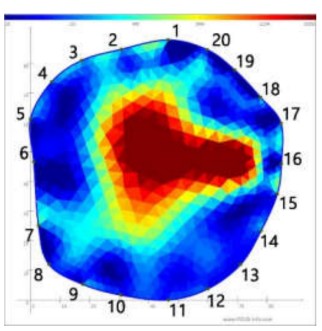 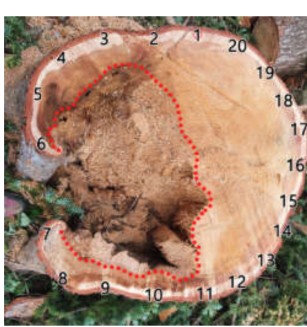

Sonic tomogram                   Electric resistance tomogram                   Disk

**(b) Tree no.4: Height of Measurement 25 cm**

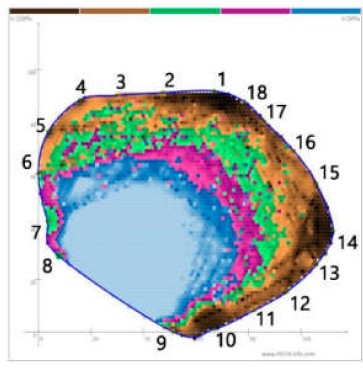 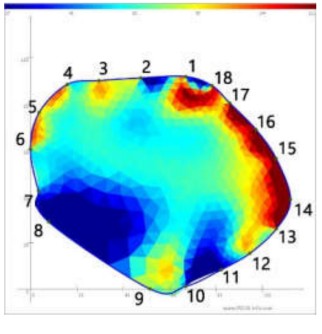 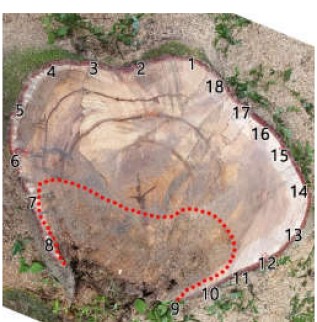

Sonic tomogram                   Electric resistance tomogram                   Disk

**Figure 3.** *Cont.*

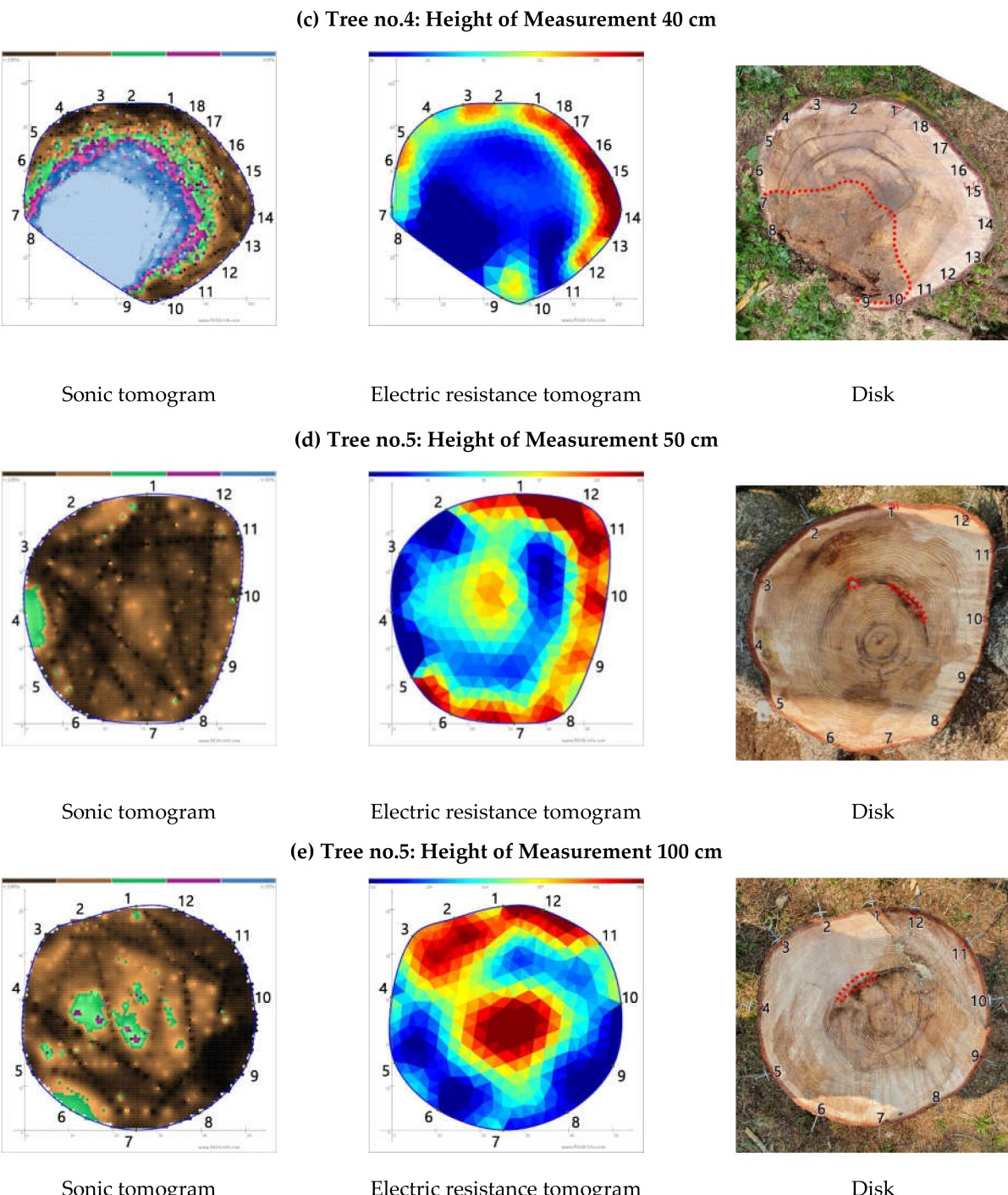

**Figure 3.** Comparison of the sonic tomograms and ERT tomograms images and the tree cross section (the red dotted lines indicate the decay area). In the sonic tomograms, brown: healthy wood, red and blue: defects, green: initial decays; in the ERT tomograms, blue: high water content, red: lower water content, green and yellow: decreasing water content.

**Table 2.** Comparison of the sonic tomograms and the tree cross sections of the compromised wood areas.

| | Compromised Wood Area (%) | | |
| --- | --- | --- | --- |
| | SoT Blue (A) | Cross Section (B) | A − B |
| (a) | 26.5 | 38.8 | −12.3 |
| (b) | 48.3 | 26.0 | 22.3 |
| (c) | 50.8 | 29.4 | 21.4 |
| (d) | 0.0 | 0.45 | −0.45 |
| (e) | 0.7 | 0.25 | 0.45 |

## 4. Discussion

Overall, SoT efficiently detected the location and size of internal defects in *A. holophylla* trees. There was no significant difference in the average defect length in wood between the results obtained by RM and SoT, and the regression analysis also showed a high explanatory power of 67%, leading to the conclusion that the two measurement results were consistent with each other.

As a result of comparing the defective area with the SoT and ERT images, the average error between the SoT image and the actual decay area in the wood section was 11%. Combining the results of the visual inspection, the tomograms seemed to reflect the trends in the location and size of the decay in the disk. The level and boundary of decay could be estimated more in detail by performing ERT on the defective area identified by SoT. In tree no. 4, when the area corresponding to the defective area in the SoT and the ERT images showed a particularly high moisture content (dark blue), severe decay and damaged sapwood were observed in the actual cross section. This shows that SoT in combination with ERT is effective in identifying the location and size of internal defects in in *A. holophylla* trees.

However, there was a tendency to overestimate the decay area compared to the actual decay area. In tree no. 4, the decay area in the tomograms was larger than the actual decay area by up to 22%. As a result of examining the cross section, the ring cracks were identified as the cause of such an error. This observation was consistent with the warning provided in the operating manual [19]. These cracks were mostly oriented in the radial direction and extended up and down in the vertical planes within the trunk, effectively cutting off linear propagation of acoustic waves, diverting them to a much longer travel path [8]. Therefore, for general trees, such as street trees, to which the invasiveness measurement method can be applied, the accuracy of the measurement can be improved by performing RM to confirm the decay and cracks identified in tomograms.

However, SoT is a highly applicable method for diagnosing internal defects in old trees, such as natural monuments and protected trees, which have high historical and cultural value and are vulnerable to physical damage. As SoT has been studied mainly in European and tropical rainforest species, and previous studies have reported differences in accuracy depending on the specific wood type for each species, continuous research on the verification of the reliability and interpretation of results according to species is required.

## 5. Conclusions

Applying nondestructive SoT to old *A. holophylla* trees, we conclude that SoT was effective in detecting the size and location of internal defects. In addition, image analysis was performed to acquire quantitative data on the internal defects of trees. However, as internal ring cracks may lead to an overestimation of defects, compared to actual defects, for a more detailed detection of the cracks and decay identified in tomograms in consideration of the characteristics of wood by tree type, the accuracy can be improved by performing RM in addition to SoT. SoT is an essential diagnostic method applicable to trees belonging to cultural heritage, such as old trees, and is effective in preventive tree management through

the monitoring of mid- to long-term changes in internal decay or cavities that are difficult to diagnose with the naked eye.

**Author Contributions:** Conceptualization; methodology; analysis; writing, J.S. (Jiwon Son); Visualization, G.L. and J.S. (Jinho Shin) All authors have read and agreed to the published version of the manuscript.

**Funding:** This work was supported by the National Research Institute of Cultural Heritage [NRICH-2105-A13F-1].

**Data Availability Statement:** Data used in this study can be requested from the corresponding author via email.

**Acknowledgments:** This work was supported by the National Research Institute of Cultural Heritage.

**Conflicts of Interest:** The authors declare no conflict of interest.

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
