# Peer review of "Reliability of Noninvasive Sonic Tomography for the Detection of Internal Defects in Old, Large Trees of Abies holophylla Maxim"

_forests, doi:10.3390/f12081131_

Round 1

Reviewer 1 Report

Comments and Suggestions for Authors

This is the review of the manuscript  (Manuscript ID: forests-1347046)

Journal: Forests

Authors: Jiwon Son, Gwanggyu Lee, Jinho Shin

Title: : Reliability of Non-invasive Sonic Tomography for the Detection of Internal Defects in Old Large Trees of Abies holophylla Maxim

Authors compare non-invasive sonic tomography with electric resistance tpmography and with disks to detect internal defects in Abies holophylla Maxim. in South Korea.

I have few comments and suggestions to authors.

Below I list specific comments:

methodical work, interesting topic

line 112 - does outer skin mean bark ??

line 124 you say that p = 0.663 but in Table 1 it is 0.661

Table 1 explain in the table caption what ID means, MP, meaning Blue next to SoT; the tables should be divided into two parts with a vertical line, which will facilitate its analysis; why the last row of the table includes Tree no., Sonic tomogram and Resistograph profile (MP) - unless it's already part of

Fig. 1, but you can't see it in pdf

Figure 1 - in the caption, explain the meaning of all the colors, e.g. brown - bark ?? and blue ?? please also give larger numbers with the sonic tomogram because then they are described in the text, and the numbers are so small that it is not known what to analyze

Figure 3 - no match between the drawing and the description (e.g. SoT in the caption and Sonic tomogram in the drawing), give larger numbers for SoT and ERT - as for disks; give the same drawing orientation for a given tree, e.g. tree 4 SoT and ERT have a different orientation than disk, which makes it difficult to analyze the results; in the caption again explain the meaning of all the colors - this is methodical work and it is very important

Data availability statement: it is worth posting the data in an open data repository, under an open license with attribution, e.g. CC BY 4.0

Author Response

Thank you for your kind comments to improve the manuscript.

-Table 1: Descriptions such as MPs were added to the footnotes of the table, and IDs were deleted, as they appeared unnecessary. And I divided the table into two parts with a vertical line.

-Figures 1 & 3 were modified

-Finally, we will review the provision of open data.

Reviewer 2 Report

„Reliability of Non-invasive Sonic Tomography for the Detection of Internal Defects in Old Large Trees of Abies holophylla Maxim” aims at verifying the effectiveness and accuracy of the sonic tomography method in detecting internal defects in trees that are not visibly by a naked eye.

The measurements were planned and performed correctly. The results are interesting. However, the paper lacks scientific explanations of the results obtained and the differences observed between the two methods compared. The manuscript lacks conclusions, which is a necessary part of a scientific article – please supplement all deficiencies to make the paper scientifically sound.

Author Response

Thank you for your kind comments to improve the manuscript.

3.2 Comparison of sonic tomograms and cross-section images:

-To supplement the comparison results of SoT and cross-section, a quantitative analysis of the defective area obtained by SoT and cross-section was added (results displayed in Table 2).

Discussion:

-An explanation about the comparison of the results obtained by SoT and those by the two other measurements was added. We further added text on the cause of errors (e.g., reasons for the overestimation of the decay area).

-Finally, a conclusions section was added to summarize the main content and considerations of our study.

Reviewer 3 Report

Dear authors
Are large circular cracks therefore detected by Picus? You write that small ones are not, is the resistograph therefore better suited in these situations? Some fungi develop this way in wood e.g. Phellinus pini in pines. For wood rot and cavities both methods are effective, right?
And what about root systems? Can root infestation of urban trees by Armillaria fungi be established? This is important because old trees growing along streets are susceptible to being blown down. This is due both to the limited space in which they can grow (near the pavement, in the street), to the compaction of the soil (lack of air) and to the fungal infestation. Which instrument do you recommend and does measurement at the base of the trunk give information about the condition of the roots?
And what about double trunks? Do we measure (and with what?) each of them separately, or the fusion point exposed to rot.
At what damage (loss of wood hardness) and surface (e.g. 2/3 of the trunk has rotted away) should the tree be removed?

L83 There was 54 MPs (measuring points) and in Tab. 1 there is 64 ID, why this difference?

L122 add space between 5.6 and cm (check the manuscript)

Author Response

Are large circular cracks therefore detected by Picus? You write that small ones are not, is the resistograph therefore better suited in these situations?

-Text mentioning that internal ring cracks are likely to lead to an overestimation of the decay and cracks by SoT and that the use of a resistograph increases the accuracy was added to the Discussion.

-A resistograph is suitable for detecting small cracks in wood for which invasive methods are applicable.

Some fungi develop this way in wood e.g. Phellinus pini in pines. For wood rot and cavities both methods are effective, right?

-A combination of both methods would be most effective.

And what about root systems? Can root infestation of urban trees by Armillaria fungi be established? Which instrument do you recommend and does measurement at the base of the trunk give information about the condition of the roots?

-There are cases in which SoT was applied to the detection of root decay, including Armillaria fungi, which is known to require further reliability verification. Although root distribution mapping using a physical probe is being actively conducted, there are few studies on the detection of the root growth state.

And what about double trunks? Do we measure (and with what?) each of them separately, or the fusion point exposed to rot.

-Gilbert et al. (2016) recommended analyzing the geometry separately if the shape is irregular, regardless of a double trunk.

At what damage (loss of wood hardness) and surface (e.g. 2/3 of the trunk has rotted away) should the tree be removed?

- There is a criterion for evaluating the necessity of removing a tree: If t/r<0.3 for a full crowned tree exposed to wind loads or if the area of the internal cavity portion accounts for approximately 70% or more (Dunster et al. 2013; Mattheck et al. 2015). However, when considering tree removal, the location of the tree and the severity of the crown should be carefully considered.

L83 There was 54 MPs (measuring points) and in Tab. 1 there is 64 ID, why this difference?

- The number of IDs is 54, which is the same as that of MPs. However, as the “IDs” seem unnecessary, they were removed from the table.

Round 2

Reviewer 2 Report

The manuscript has been supplemented and corrected properly. I can recommend it for publication.